# Optical Temperature Sensing of YbNbO$_4$:Er$^{3+}$ Phosphors Synthesized by Hydrothermal Method

**Heming Ji** [1]**, Xunze Tang** [1]**, Haiyan Zhang** [1]**, Xiaolong Li** [2] **and Yannan Qian** [1],*

1. Guangdong Provincial Key Laboratory of Functional Soft Condensed Matter, School of Materials and Energy, Guangdong University of Technology, Guangzhou 510006, China; hemingj@mail2.gdut.edu.cn (H.J.); yangz@mail2.gdut.edu.cn (X.T.); hyzhang@gdut.edu.cn (H.Z.)
2. School of Mechanical and Electrical Engineering, Yunnan Agricultural University, Kunming 650201, China; 2015028@ynau.edu.cn
* Correspondence: qianyannan@gdut.edu.cn

**Abstract:** The novel YbNbO$_4$:Er$^{3+}$ phosphors were firstly synthesized through the hydrothermal method by adding LiOH·H$_2$O as flux in the H$_2$O/EG system. YbNbO$_4$:Er$^{3+}$ phosphors showed the agglomerated irregular polygons coexisting with some tiny grains. XRD and Raman spectra were measured to understand the phase structure and the crystal growth mechanism of YbNbO$_4$:Er$^{3+}$ phosphors. The upconversion (UC) emission spectra, the pump power dependency and UC mechanism were studied under 980 nm excitation. Based on the fluorescence intensity ratio technique, YbNbO$_4$:Er$^{3+}$ exhibited the maximum sensor sensitivity of 0.00712 K$^{-1}$ at 220 K, providing a promising application in optical low-temperature sensors.

**Keywords:** optical temperature sensing; YbNbO$_4$:Er$^{3+}$ phosphor; fluorescence intensity ratio





## 1. Introduction

Non-contact optical thermometry based on fluorescence intensity ratio (FIR) technique is playing a key role in applications for harsh environments, such as high-intensity electromagnetic fields, fire detection and electrical power stations due to its excellent properties of fast response and anti-interference, high-spatial resolution and superior accuracy [1,2]. The FIR technique, which utilizes the temperature-dependent emission intensity from the two thermally coupled energy levels (TCLs) of rare-earth (RE$^{3+}$) ions, is independent of spectrum losses and fluctuations of exciting intensity [3,4]. Er$^{3+}$ ion is an ideal candidate for optical temperature sensor because its energy gap ($\Delta E$) between $^2$H$_{11/2}$ and $^4$S$_{3/2}$ states is about 700–800 cm$^{-1}$, satisfying the requirement for locating in the range of $200 \leq \Delta E \leq 2000$ cm$^{-1}$ [5,6]. To our knowledge, much attention has been focused on the non-contact optical thermometry in the high-temperature circumstance [7–9]. Ye reported the maximum absolute sensitivity value of 0.0552 K$^{-1}$ and the relative sensitivity value of 1.49% K$^{-1}$ recorded from 294 to 573 K in Ba$_3$Y$_4$O$_9$:Ho$^{3+}$/Tm$^{3+}$/Yb$^{3+}$ phosphor [7]. It has been reported by Chen that $\beta$-NaYF$_4$:Yb$^{3+}$/Er$^{3+}$ microcrystal exhibited the maximum sensitivity of 0.0037 K$^{-1}$ at 508 K in the temperature region of 298–653 K [8]. YPO$_4$:Yb$^{3+}$/Ho$^{3+}$/Tm$^{3+}$ submicro-plates synthesized by Lu possessed a high temperature sensitivity of 0.0885 K$^{-1}$ at 563 K according to the thermo-responsive FIR of Ho$^{3+}$ ($^5$F$_5 \rightarrow$ $^5$I$_8$) to Tm$^{3+}$ ($^3$F$_3 \rightarrow$ $^3$H$_6$) emissions [9]. However, there are few reports on optical temperature sensors in the cryogenic region below 298 K. Kaczmarek and Deun showed that LaF$_3$:Yb$^{3+}$/Er$^{3+}$ nanoparticles displayed a remarkably high relative sensitivity of 0.6092% K$^{-1}$ at 15 K [10].

RE$^{3+}$ ions doped lanthanide niobates (LnNbO$_4$, Ln = La, Gd, Tb and Y, et al.) have garnered a tremendous amount of attention since LnNbO$_4$ has the advanced physical properties, such as the good thermal and chemical stability, a wide transparency range, electro-optical, the high dielectric constants and nonlinear optics [11–13]. Zhou reported the



colorful visible emissions in $LaNbO_4$:$Yb^{3+}$/$Er^{3+}$/$Ho^{3+}$ materials under 980 nm excitation due to the low symmetry of $LaNbO_4$ [14]. It has been reported by Carmo that adjusting the concentrations of $Tm^{3+}$ and $Yb^{3+}$ ions was responsible for producing the white up-conversion (UC) emission in $YNbO_4$:$Er^{3+}$/$Tm^{3+}$/$Yb^{3+}$ phosphor [15]. $YNbO_4$:$Eu^{3+}$/$Er^{3+}$ phosphors synthesized by Yin displayed a relative sensitivity of 0.0088 $K^{-1}$ at 303 K [16]. However, there was only one report for $YbNbO_4$ thin film prepared by sol-gel method [17]. Therefore, it is desirable to discuss the phase evolution and optical low-temperature sensing properties of $RE^{3+}$ ions in $YbNbO_4$ phosphors. Furthermore, the $YbNbO_4$:$Er^{3+}$ phosphors will provide an opportunity to synthesize the flexible coatings for temperature sensing and photothermal conversion. For example, Gonçalves and Ferreira reported that the $GeO_2$-$Ta_2O_5$:$Er^{3+}$/$Yb^{3+}$ particles were dispersed in poly(methyl methacrylate) (PMMA) to form the plastic free-standing films. The high-quantum-yield UC $Er^{3+}$/$Yb^{3+}$-organic-inorganic hybrid coatings exhibited a relative thermal sensitivity of similar to 1.1% $K^{-1}$ at 300 K [18].

In this work, $YbNbO_4$:$Er^{3+}$ phosphors are synthesized by hydrothermal method in $H_2O$/EG solution system for the first time. The phase evolution, optical characteristics and optical low-temperature sensing of $Er^{3+}$ ions in $YbNbO_4$ phosphors are discussed.

## 2. Materials and Methods

In a typical hydrothermal method for the synthesis of $YbNbO_4$:$x$ mol% $Er^{3+}$ ($x = 0.1$ and 0.2), 1.5 mmol of $Yb(NO_3)_3 \cdot 5H_2O$ (99.9%), 3.6 mmol of $Nb_2O_5$ (99.99%), 0.1 mmol/0.2 mmol of $Er(NO_3)_3 \cdot 5H_2O$ (99.9%) and 7.0 mmol of $LiOH \cdot H_2O$ were dissolved in $H_2O$/EG mixture solution (volume ratio of 40 mL:40 mL). Here, an excess of $Nb_2O_5$ was used due to its poor solubility. Then, the above solution was heated at 270 °C for 12 h. After cooling naturally to ambient temperature, the white precipitates were centrifuged and washed with deionized water and ethanol three times, and dried at 70 °C. Finally, $YbNbO_4$:0.1 mol% $Er^{3+}$ and $YbNbO_4$:0.2 mol% $Er^{3+}$ phosphors, named as YNE-1 and YNE-2, respectively, were obtained through annealed at 900 °C for 2 h.

The powder X-ray diffraction (XRD) spectra were measured by using a powder diffractometer equipped with Cu K$\alpha$ radiation source (40 kV, 30mA, $\lambda = 1.5406$ Å, Bruker AXS D8-Advance, Karlsruhe, Germany). Scanning electron microscopy (SEM, Hitachi SU8010, Tokyo, Japan) was used to observe the morphology. Raman spectra were studied by a micro co-focal Raman spectrometer (Horiba LabRAM HR Evolution, Longjumeau, France). Under a 980 nm laser excitation, the temperature-dependent UC emission spectra were measured by a fluorescence spectrometer system (Zolix, Beijing, China) equipped with a temperature controller (Lake Shore Model 336, Westerville, OH, USA).

## 3. Results

The XRD patterns shown in Figure 1a indicate that the main diffraction peaks of YNE-1 and YNE-2 phosphors can be indexed to monoclinic phase $YbNbO_4$ (JCPDS Card No. 81-1976) with an impurity phase $Yb_2O_3$ observed from its 2$\theta$ reflection at 29.7° (JCPDS Card No. 43-1037). An impurity phase $Nb_2O_5$ (JCPDS Card No. 72-1121) is observed in YNE-2.

Figure 1b shows the vibrations associated to Raman scattering for YNE-1 and YNE-2 phosphors. Raman bands around 117, 183, 303, 316, 334, 418, 444 and 816 $cm^{-1}$ are agreed with the literature data reported for the $YbNbO_4$ [19]. The Raman peaks at 418 $cm^{-1}$ and 334/816 $cm^{-1}$ are assigned to Nb-O anti-symmetric and symmetric modes of $NbO_4$ tetrahedral structure, respectively, meaning that YNE-1 and YNE-2 have a regular $NbO_4$ tetrahedron with no interactions and distortions [20]. Additionally, $YbNbO_4$:$Er^{3+}$ phosphors here also possess the similar structure to H-$Nb_2O_5$ due to the appearance of Raman bands at 238, 630, 678 and 992 $cm^{-1}$ [21]. The Raman peaks observed at 540, 834, 901 and 935 $cm^{-1}$ represent the existent of the impurity phase $Nb_2O_5$ [22–24], and Raman bands at 470 $cm^{-1}$ is assigned to the phase $Yb_2O_3$ [25]. The phonon energies below 300 $cm^{-1}$, including 135, 155 and 276 $cm^{-1}$, are assigned to external vibrations [26]. Referring to XRD

and Raman spectra, the formation evolution of $YbNbO_4:Er^{3+}$ could be understood by the following equations [20,27]:

In the dissolution-precipitation processes:

$$3\,Nb_2O_5 + 8\,OH^- \rightarrow Nb_6O_{19}{}^{8-} + H_2O \tag{1}$$

$$Nb_6O_{19}{}^{8-} + 34\,OH^- \rightarrow 6\,NbO_6{}^{7-} + 17H_2O \tag{2}$$

$$NbO_6{}^{7-} + Li^+ + 3\,H_2O \rightarrow LiNbO_3 + 6\,OH^- \tag{3}$$

$$LiNbO_3 + Yb^{3+} + 2\,OH^- \rightarrow YbNbO_4 + Li^+ + H_2O \tag{4}$$

$$Yb^{3+} + 3\,OH^- \rightarrow Yb(OH)_3 \tag{5}$$

In the calcination process:

$$2\,Yb(OH)_3 = Yb_2O_3 + 3H_2O \tag{6}$$

In a dissolution-precipitation process, due to the poor solubility and weak acidic of $Nb_2O_5$, $LiOH \cdot H_2O$ is used as the flux. At the initial stage, $Nb_2O_5$ is dissolved into $Nb_6O_{19}{}^{8-}$ ions based on a similar neutralization reaction between an acid and a base (Equation (1)). Then, $Nb_6O_{19}{}^{8-}$ furtherly reacts with more $OH^-$ to form single octahedron $NbO_6{}^{7-}$ anions via complex transformations (Equation (2)). In a supersaturated medium (Equation (3)), the phase $LiNbO_3$ is occurred after producing the tiny crystalline nucleation. Finally, $YbNbO_4$ is generated through the exchange reaction between $Yb^{3+}$ and $Li^+$ ions based on Equation (4). It is inevitable that $Yb^{3+}$ ions would react with $OH^-$ in the dissolution-precipitation process (Equation (5)). After calcinating, the $Yb_2O_3$ phase is formed by the decomposition reaction of $Yb(OH)_3$ [28]. The phase $Nb_2O_5$ appeared in YNE-2 (Figure 1a) is caused by the suppressed exchange reaction process between $Yb^{3+}$ and $Li^+$ ions, since the enhanced $Er^{3+}$ ions consume $OH^-$ ions. It is speculated that the impurities $Nb_2O_5$ and $Yb_2O_3$ would produce the new defect centers in $YbNbO_4$ host matrix, which may decrease the UC emissions under 980 nm excitation.

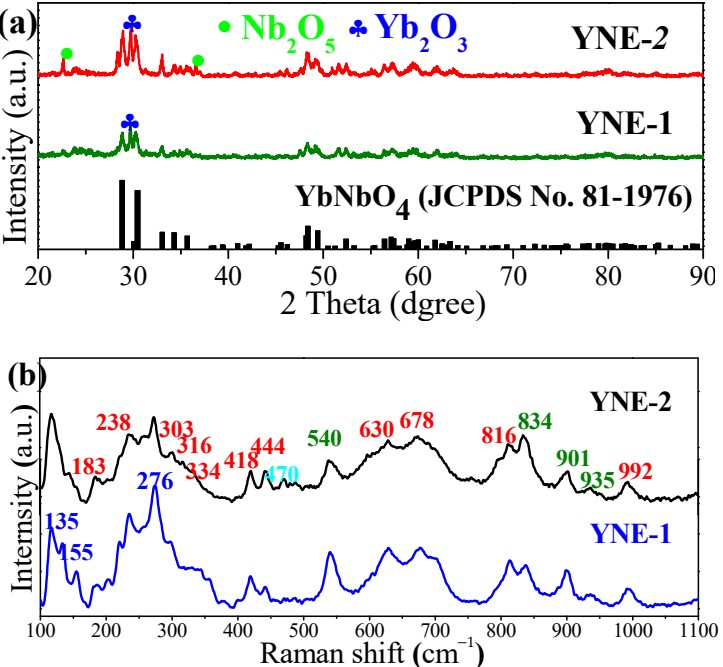

**Figure 1.** YNE-1 and YNE-2 phosphors; (**a**) XRD patterns; (**b**) Raman spectra.

SEM images shown in Figure 2 display YNE-1 and YNE-2 phosphors are composed of the agglomerated irregular polygons with an average diameter of 0.5–1.0 μm and tiny grains. It is possible that the non-uniform tiny grains may be the impurity $Yb_2O_3$ particles.

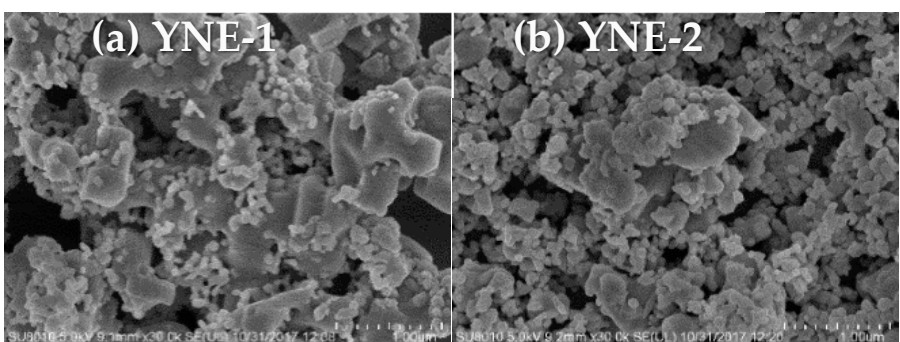

**Figure 2.** SEM images of $YbNbO_4:Er^{3+}$ (**a**)YNE-1, (**b**)YNE-2.

Figure 3a shows the UC emission spectra of $YbNbO_4:Er^{3+}$ phosphors under 980 nm excitation. The two green UC emissions at 530/558 nm and a red UC emission at 672 nm are attributed to the $^2H_{11/2}/^4S_{3/2} \rightarrow {}^4I_{15/2}$ and $^4F_{9/2} \rightarrow {}^4I_{15/2}$ transitions of $Er^{3+}$ ions, respectively [29]. It is obvious that both green and red UC emissions decrease with the increasing concentrations of $Er^{3+}$ ions in YNE-2 phosphor. Combination of XRD, Raman spectra and UC emission spectra, there are no studies on $YbNbO_4$ phosphors doped with higher concentration of $Er^{3+}$ ions. This is because the increasing $Er^{3+}$ content leads to the appearance of the impurity of $Nb_2O_5$ and the reduced UC emissions.

To understand UC mechanisms, the intensity of fluorescence ($I_f$) is measured as a function of the pump power ($P$). Log–Log plots of pump power dependence in $YbNbO_4:Er^{3+}$ phosphors are shown in Figure 3b. For an unsaturated UC process, the number of pump photons ($n$) required to populate the emitting state can be obtained via the formula: $I_f \propto P^n$ [30]. As for YNE-1 and YNE-2, respectively, the slope values of the green UC emissions are fitted to be 2.25 and 2.34, and the red UC emissions yield $n = 1.83$ and 1.94, indicating that at least two 980 nm laser photons are needed to populate both green and red UC emissions [31].

Figure 3c depicts schematically possible UC mechanisms of $YbNbO_4:Er^{3+}$ phosphors under 980 nm excitation. Previous studies on $LnNbO_4$ indicated that $Ln^{3+}$ ions acted not only as one component of host materials but also as the sensitizer to absorb laser excitation and transfer its energy to active ions. For example, The $Gd^{3+}$ ion, a part of $GdNbO_4$ self-activated compound, could transfer its absorbed energy to the state $^1D_2$ of $Tm^{3+}$ and the state $^4F_9$ of $Dy^{3+}$ ions, leading to the blue and green emissions under UV light excitation [32]. $TbNbO_4$ emitted the green emission at 549 nm arising from the $^5D_4 \rightarrow {}^7F_5$ transition of $Tb^{3+}$ ions [33]. Therefore, 980 nm laser excitation of $Yb^{3+}$ ions is only considered here, since $Yb^{3+}$ ion has a much larger absorption cross section and could efficiently transfer its absorbed energy to $Er^{3+}$ ions [34]. As illustrated in Figure 3c, the $^4F_{7/2}$ state of $Er^{3+}$ ions is populated by the energy transition (ET) processes of ET1: $^4I_{15/2} (Er^{3+}) + {}^2F_{5/2} (Yb^{3+}) \rightarrow {}^4I_{11/2} (Er^{3+}) + {}^2F_{7/2} (Yb^{3+})$ and ET2: $^4I_{11/2} (Er^{3+}) + {}^2F_{5/2} (Yb^{3+}) \rightarrow {}^4F_{7/2} (Er^{3+}) + {}^2F_{7/2} (Yb^{3+})$ [35]. Subsequently, the $Er^{3+}$ ions on the $^4F_{7/2}$ state nonradiatively relax to the $^2H_{11/2}/^4S_{3/2}$ states, which decay radiatively to the $^4I_{15/2}$ state, producing the green UC emissions. The $Er^{3+}$ ions at $^4F_{9/2}$ state, which are populated by ET3 process of $^2F_{5/2} (Yb^{3+}) + {}^4I_{13/2} (Er^{3+}) \rightarrow {}^2F_{7/2} (Yb^{3+}) + {}^4F_{9/2} (Er^{3+})$, radiatively depopulate to the $^4I_{15/2}$ state, generating the red UC emission. It is proposed that the energy back-transfer (EBT) process of $^4S_{3/2} (Er^{3+}) + {}^2F_{7/2} (Yb^{3+}) \rightarrow {}^4I_{13/2} (Er^{3+}) + {}^2F_{5/2} (Yb^{3+})$ may occur [36]. An increase in the $Er^{3+}$ concentrations leads to a shortened distance between $Yb^{3+}$-$Er^{3+}$ pairs and a fast EBT process because the rate of ET process is inversely proportional to the distance between two neighboring ions. Consequently, the reduction of green and red emissions in YNE-2 arises from more efficient EBT process (see Figure 3c).

This is an indication that the optical quenching exists in YbNbO$_4$:Er$^{3+}$ phosphors at high doping concentration of Er$^{3+}$ ions.

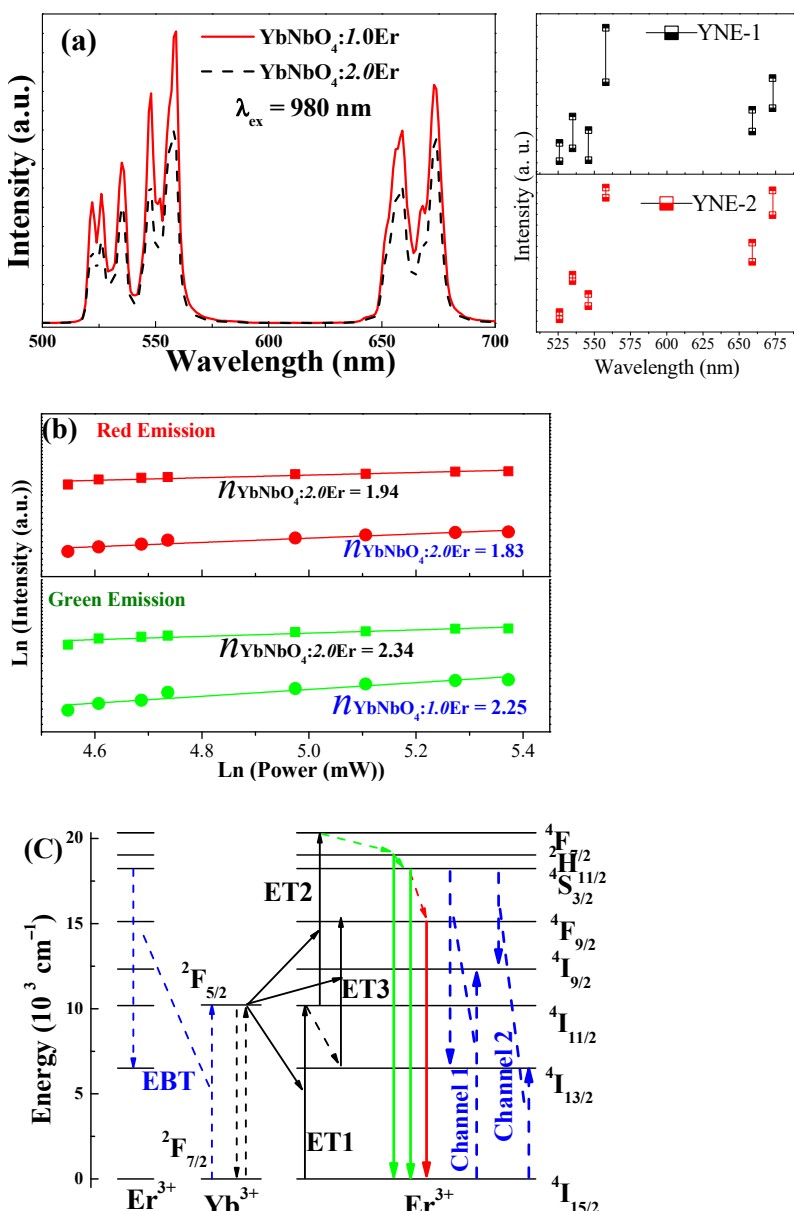

**Figure 3.** (**a**) Upconversion (UC) emission spectra; (**b**) pump power dependency; (**c**) UC mechanism of YbNbO$_4$:Er$^{3+}$ under 980 nm excitation. Error bars represent the standard deviations from three repeated measurements.

Figure 4 shows temperature-dependent on green emissions at 530 nm/558 nm of YNE-1 phosphor in the temperature range of 120–280 K under 980 nm excitation. As the temperature increases (Figure 4a), the intensity of the green emission at 530 nm is observed to increase with respect to the emission at 558 nm, and the red UC emission at 672 nm increases. As illustrated in Figure 4b, the intensity ratio of the overall green to red emission (Ratio of green and red) is increased with increasing temperature. The international commission on illumination (CIE) chromaticity diagram used to reflect the true color of luminescences at different temperatures is shown in Figure 4c. Based on 1931 CIE chromaticity theory [37], the color coordinates (*x, y*) of YNE-1 phosphor are calculated to be (0.40, 0.59), (0.39, 0.60), (0.38, 0.61), (0.37, 0.62), (0.37, 0.62), (0.35, 0.63), (0.34, 0.64), (0.33, 0.65) and (0.32, 0.66), respectively, from the temperature ranging from 120 to 280 K.

Therefore, the observed gradual change of color tone from yellow to green region implies that YbNbO$_4$:Er$^{3+}$ may have an ability of the temperature-dependent color tuning property.

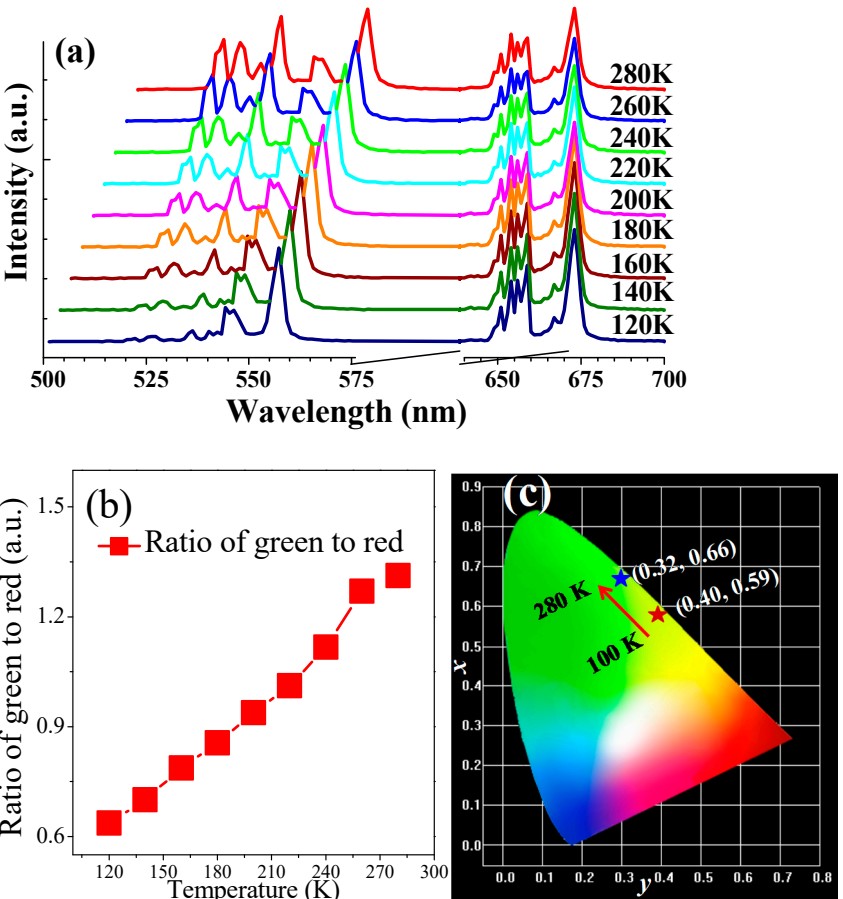

**Figure 4.** The temperature-dependent behaviors of YNE-1 phosphor measured from 120 to 280 K under 980 nm excitation: (**a**) temperature-dependent UC emission spectra; (**b**) intensities ratio of green to red; (**c**) the CIE chromaticity coordinates.

Since the populations of the $^2$H$_{11/2}$ and $^4$S$_{3/2}$ TCLs of Er$^{3+}$ ion obey the Boltzmann thermal equilibrium, the relationship between FIR, which is the ratio of $I_{530}$ to $I_{558}$, and the corresponding temperature is expressed as the following equation [38,39]:

$$\text{FIR} = \frac{I_{530}}{I_{558}} = C \exp\left(-\frac{-\Delta E}{k_B T}\right) \tag{7}$$

Where $I_{530}$ and $I_{558}$ represent the intensities of green emissions around 530 and 558 nm, respectively; $\Delta E$ means an energy gap between the two TCLs; k$_B$, $T$ and $C$ are the Boltzmann constant, the absolute temperature and the constant, respectively.

Figure 5a illustrates FIR of $I_{530}$ and $I_{558}$ as a function of the temperature in the range of 120~280 K, giving the slope value of $\Delta E/k_B$ is fitted to be about −454.96 cm$^{-1}$*K in YNE-1. The behavior that the fitted $\Delta E$ of 318 cm$^{-1}$ is much smaller than the experimental $\Delta E$ of 723 cm$^{-1}$ determined from Figure 3a may be attributed to the nonradiative cross-relaxation channels (Channel 1: $^4$S$_{3/2}$ + $^4$I$_{15/2}$ → $^4$I$_{13/2}$ + $^4$I$_{9/2}$ and Channel 2: $^4$S$_{3/2}$ + $^4$I$_{15/2}$ → $^4$I$_{9/2}$ + $^4$I$_{13/2}$) (see Figure 3c).

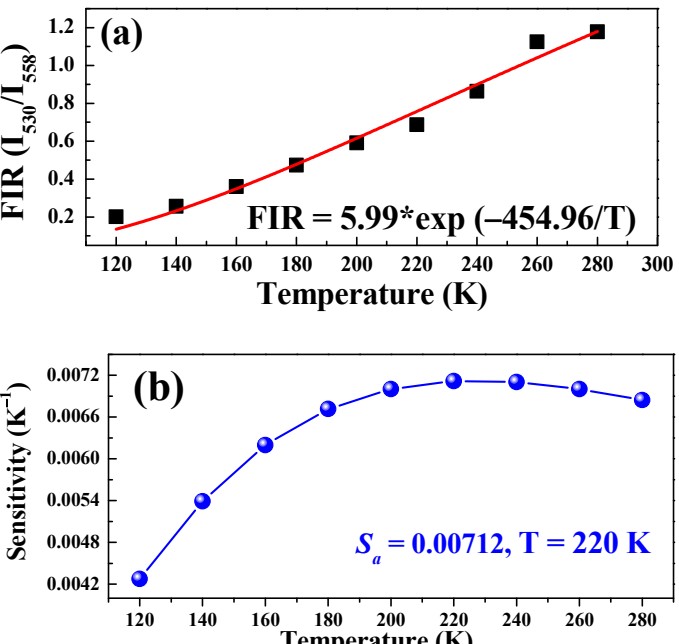

**Figure 5.** (**a**) Fitted plots of FIR ($I_{530}/I_{558}$) versus temperature; (**b**) $S_a$ as a function of temperature of YNE-1.0.

The absolute sensor sensitivity $S_a$ plays an important role in determining the practical application of optical thermal sensing, which can be obtained by [40]:

$$S_a = \frac{d\,FIR}{dT} = FIR\frac{\Delta E}{k_B T^2} \tag{8}$$

In Figure 5b, the sensor sensitivity $S_a$ calculated by means of Equation (8) decreases at elevated temperature. It has been found that the maximum $S_a$ ($S_{max}$) of 0.00712 K$^{-1}$ at 220 K in YNE-1 phosphor is higher than $S_{max}$ of 0.0053 K$^{-1}$ at 350 K in Gd$_2$MoO$_6$:Yb$^{3+}$/Er$^{3+}$ [41], $S_{max}$ of 0.0053 K$^{-1}$ at 93 K in BaCaTiO$_3$:Ho$^{3+}$/Yb$^{3+}$ [42], $S_{max}$ of 0.0044 K$^{-1}$ at 637 K in NaYF$_4$:Er$^{3+}$/Yb$^{3+}$ [43] and $S_{max}$ of 0.0037 K$^{-1}$ at 508 K in $\beta$-NaYF$_4$:Er$^{3+}$/Yb$^{3+}$ phosphor [44], and is comparable to $S_{max}$ of 0.0073 K$^{-1}$ at a temperature of 473 K in YNbO$_4$:Yb$^{3+}$/Er$^{3+}$ phosphor [45]. Therefore, the $S_{max}$ of 0.00712 K in YbNbO$_4$:Er$^{3+}$ phosphor is as large as possible to provide a promising application for monitoring the low temperature.

## 4. Conclusions

In summary, the monoclinic YbNbO$_4$:Er$^{3+}$ phosphors composed of the agglomerated irregular polygons and tiny grains are successfully synthesized for low-temperature optical sensors. Here, Yb$^{3+}$ ions, one component of YbNbO$_4$ host matrix, could transfer their absorbed energy to excite Er$^{3+}$ ions to the $^2$H$_{11/2}$/$^4$S$_{3/2}$ and $^4$F$_{9/2}$ emitting states. The elevating temperature leads to a multicolor change from yellow to green in YNE-1. The reduced green and red UC emissions caused by the increased concentrations of Er$^{3+}$ ions are resulted from the fast EBT process of $^4$S$_{3/2}$ (Er$^{3+}$) + $^2$F$_{7/2}$ (Yb$^{3+}$) → $^4$I$_{13/2}$ (Er$^{3+}$) + $^2$F$_{5/2}$ (Yb$^{3+}$). In the low temperature range, a high maximum sensor sensitivity of 0.00712 K$^{-1}$ at 220 K is achieved in YNE-1, contributing a feasible and expansible way to further survey the sensitivity of optical temperature sensor and promote its applications.

**Author Contributions:** Conceptualization, X.L. and Y.Q.; data curation, X.T.; writing—original draft preparation, H.J.; writing—review and editing, H.Z. and Y.Q. All authors have read and agreed to the published version of the manuscript.

**Funding:** This research was supported by Guangdong Natural Science Funds for Distinguished young scholar (No.:2015A030306041), the tip-top Scientific and Technical Innovative Youth Talents of

Guangdong special support program (No.:2015TQ01N060), the Major Foundation of Guangzhou Science and Technology (No.:201804020005).

**Institutional Review Board Statement:** Not applicable.

**Informed Consent Statement:** Not applicable.

**Data Availability Statement:** Data sharing not applicable.

**Conflicts of Interest:** The authors declare no conflict of interest.

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
