# Peer review of "Optical Temperature Sensing of YbNbO4:Er3+ Phosphors Synthesized by Hydrothermal Method"

_coatings, doi:10.3390/coatings11040383_

Round 1

Reviewer 1 Report

This work introduces the use of Erbium-compound as up-conversion host for sensing the temperature change. It worth to be published, but there are some minor comments needed to be addressed first:

  • Did the authors try to coat the synthesized material on a substrate, or use it AS IS? As the scope of the journal is related to Coatings so it be preferred to clarify this point.
  • Why did not the authors try higher doping ratios of erbium? If you detected optical quenching of the emitted optical signals, so it has to be mentioned in the manuscript.
  • The sensing methodology depends on the ratio between green/red emission. Most of sensors depend on only relative change of one emission with temperature. So, could you add a reference from literature to support the method that you used?
  • The emission intensities and its relative changes have to be included with error bars.
  • In the synthesis procedure, why did not the author try higher calcination temperatures? From other literature that could help in forming more metastable states in upconversion process which leads to higher optical efficiency.
  • Figure 4 a needs to fix the x-axis, and the axes mentioned in Figure 4c are not clear.

Reviewer 2 Report

The authors presented research on a novel phosphor for temperature sensing YbNbO4 doped with Er3+. The upconversion luminescence exhibited substantial thermal sensitivity which warrants its application in a wide temperature range.

The manuscript is properly structured, written, and formatted. There are only few minor issues that I want authors to address. I recommend the article be published after a minor revision.

  • Please comment in one sentence on the possibility of the impurities (Nb2O5, Yb2O3) interference with the spectroscopic results and thermal sensing properties
  • 4. Contrary to the statement in the text, the red component seems to decrease with increasing temperature.
  • ΔE/kB should be in unit cm-1*K
  • 4. Caption a): The is only one plot of FIR presented
  • I would encourage the authors to differentiate between the absolute (Sa) and relative sensitivity (Sr) when comparing the results to those in the literature.
